# Parent Involvement in Diet or Physical Activity Interventions to Treat or Prevent Childhood Obesity: An Umbrella Review

**DOI:** 10.3390/nu13093227

**Published:** 2021-09-16

**Authors:** Emily J. Tomayko, Alison Tovar, Nurgul Fitzgerald, Carol L. Howe, Melanie D. Hingle, Michael P. Murphy, Henna Muzaffar, Scott B. Going, Laura Hubbs-Tait

**Affiliations:** 1Center for American Indian and Rural Health Equity, Montana State University, Bozeman, MT 59718, USA; 2Department of Nutrition and Food Sciences, University of Rhode Island, Kingston, RI 02881, USA; alison_tovar@uri.edu; 3Department of Nutritional Sciences, Rutgers, The State University of New Jersey, New Brunswick, NJ 08901, USA; nurgul.fitzgerald@rutgers.edu; 4Health Sciences Library, University of Arizona, Tucson, AZ 85721, USA; howca@arizona.edu; 5School of Nutritional Sciences and Wellness, College of Agriculture & Life Sciences, University of Arizona, Tucson, AZ 85721, USA; hinglem@arizona.edu (M.D.H.); going@arizona.edu (S.B.G.); 6College of Public Health and Human Sciences, Oregon State University, Corvallis, OR 97331, USA; murphym6@oregonstate.edu; 7College of Health and Human Sciences, Northern Illinois University, DeKalb, IL 60115, USA; hmuzaffar@niu.edu; 8Department of Human Development and Family Science, College of Education and Human Sciences, Oklahoma State University, Stillwater, OK 74078, USA; laura.hubbs@okstate.edu

**Keywords:** childhood obesity, nutrition, physical activity, parents, treatment, prevention, interventions

## Abstract

Parents substantially influence children’s diet and physical activity behaviors, which consequently impact childhood obesity risk. Given this influence of parents, the objective of this umbrella review was to synthesize evidence on effects of parent involvement in diet and physical activity treatment and prevention interventions on obesity risk among children aged 3–12 years old. Ovid/MEDLINE, Elsevier/Embase, Wiley/Cochrane Library, Clarivate/Web of Science, EBSCO/CINAHL, EBSCO/PsycInfo, and Epistemonikos.org were searched from their inception through January 2020. Abstract screening, full-text review, quality assessment, and data extraction were conducted independently by at least two authors. Systematic reviews and meta-analyses of diet and physical activity interventions that described parent involvement, included a comparator/control, and measured child weight/weight status as a primary outcome among children aged 3–12 years old were included. Data were extracted at the level of the systematic review/meta-analysis, and findings were narratively synthesized. Of 4158 references identified, 14 systematic reviews and/or meta-analyses (eight treatment focused and six prevention focused) were included and ranged in quality from very low to very high. Our findings support the inclusion of a parent component in both treatment and prevention interventions to improve child weight/weight status outcomes. Of note, all prevention-focused reviews included a school-based component. Evidence to define optimal parent involvement type and duration and to define the best methods of involving parents across multiple environments (e.g., home, preschool, school) was inadequate and warrants further research. PROSPERO registration: CRD42018095360.

## 1. Introduction

Childhood overweight and obesity remain at alarmingly high levels despite extensive intervention efforts [1]. Childhood obesity is a recognized risk factor for type 2 diabetes, cardiovascular diseases, respiratory diseases, musculoskeletal complications, poor health-related quality of life, and negative emotional health [2,3,4,5]. Because childhood obesity tracks into adolescence and adulthood [6,7,8], early treatment and prevention strategies are critical.

The development of overweight and obesity is complex given the biological, social, environmental, cultural, and behavioral causes [9]. Critical behaviors involved in the development of obesity include unhealthy diet and eating behaviors, low levels of physical activity, and a sedentary lifestyle [10,11]. Parents and other caregivers (herein referred to collectively as “parents”) significantly influence children’s diet, physical activity, and other health behaviors [12,13,14,15]. Parent-level factors that influence child behaviors include food purchasing and meal preparation choices; parenting style; and knowledge, attitudes, and behaviors around food, physical activity, and health [16,17,18,19,20]. These influences are particularly important within the family home environment. Many children also spend significant time in environments outside of the home, such as childcare, preschool, and/or school. For this reason, considering parent involvement as part of interventions across these various settings is recommended to increase impact on child obesity and related factors [21,22,23,24] in alignment with socioecological models of child obesity [25].

Umbrella reviews are well suited to provide a structured, evidence-based context in which various prevention and treatment options can be compared and contrasted [26,27,28]. Two previous umbrella reviews have demonstrated the effectiveness of lifestyle interventions for the treatment of child and adolescent obesity [29,30], with one of the reviews focused on family-based treatment interventions [30]. Prevention-focused umbrella reviews are inconsistent about whether interventions are effective for preventing child and adolescent obesity [24,31,32,33]. There are differences in age ranges of the systematic reviews within those umbrella reviews, which may explain some of the inconsistency in findings. While another umbrella review provided synthesis of both prevention and treatment approaches to adolescent and child obesity [34], it did not focus on the role of parent involvement. To our knowledge, no prior umbrella review has addressed both treatment and prevention meta-analyses and systematic reviews of parent involvement in child obesity interventions and focused specifically on children aged 3–12 years old.

This umbrella review addresses a gap in the literature by synthesizing evidence on parent involvement in diet and/or physical activity treatment and prevention programs on obesity risk among children aged 3–12 years old. The age range of 3–12 years old was targeted in the present umbrella review for multiple reasons: the onset of preschool programs at age 3; transition to adolescence around age 12; and the different role of parents during this period versus before and after it, such as shifts in parenting linked to changes in child language and cognition between ages 2 and 3 [35,36,37] and differences in effective parenting of children versus adolescents. It is particularly important to recognize that weight for length may be measured rather than weight for height up to 3 years of age [38], whereas weight for height is measured thereafter.

## 2. Materials and Methods

The methodology for this umbrella review adhered to protocol CRD42018095360 registered with the International Prospective Register of Systematic Reviews: http://www.crd.york.ac.uk/PROSPERO/display_record.php?ID=CRD42018095360, accessed on 14 September 2021). All authors made substantial contributions to the development and/or refinement of the study protocol. Amendments were made in 2020 to clarify language and update the study team and progress log; no substantive changes were made to any of the sections. The methodology also adhered to the guidelines in the Preferred Reporting Items for Systematic Reviews and Meta-Analyses (PRISMA) [39].

### 2.1. Search Strategies and Eligibility Criteria

Using both controlled vocabulary terms (e.g., MeSH, Emtree) and keywords, a medical librarian (C.L.H.) searched the following databases from the dates of their inception to 21 January 2020: Ovid/MEDLINE; Elsevier/Embase; Wiley/Cochrane Library; Clarivate/Web of Science; EBSCO/Cumulative Index of Nursing and Allied Health Literature; EBSCO/PsycInfo; and Epistemonikos.org. Filters were applied to identify systematic reviews (SRs) and meta-analyses (MAs) and English language. Search strategies are available in Appendix A.

Included reviews were SRs or MAs in which the population of interest was children aged 3–12 years old, or data for this age group were extractable from a larger age range (e.g., a sub-analysis within the range of interest was conducted). Interventions from included primary studies within the SRs/MAs had to focus on treatment or prevention of childhood obesity, provide information about parent involvement, and include a nutrition and/or physical activity component. Primary studies within the SRs/MAs had to feature a comparator, such as a control group, comparison group, and/or alternative treatment group. The primary outcome of primary studies within the SRs/MAs had to be a measure of child weight or weight status (e.g., body mass index [BMI], BMI z-score, BMI percentile, weight). Changes in physical activity, diet, or other health behaviors could be included as secondary outcomes.

Exclusion criteria were consistent with inclusion criteria. All reviews that were not SRs or MAs were excluded. All SRs/MAs or subsets of extractable data within SRs/MAs that did not meet one or more of the inclusion criteria above were excluded. Reviews were excluded if the only parent component of the primary studies was provision of informed consent or completion of questionnaires. Conference abstracts, proceedings, dissertations, letters, commentaries, and opinion pieces were excluded. SRs/MAs not available in English were also excluded. All studies excluded during full-text review and reasons for exclusion are available in Appendix A.

### 2.2. Study Selection

Records identified through the database searches were exported to EndNote X9 (Clarivate Analytics, Philadelphia, PA, USA) for deduplication and pre-screening. A medical librarian (C.L.H.) pre-screened initial results, removing conference abstracts, opinion articles, publications that had been withdrawn, and non-SRs/MAs. Two authors (E.J.T., M.P.M.) independently screened all titles and abstracts of remaining references for topic relevance. Disagreements were resolved by consensus or consultation with a third author (M.D.H./L.H.-T.), when needed. Two authors (E.J.T., M.P.M.) screened the full text of those publications selected during title/abstract review, resolving disagreements by consensus or consultation with a third author (M.D.H./L.H.-T.).

### 2.3. Quality Assessment

Two authors (E.J.T., L.H.-T.) independently assessed studies for quality using the AMSTAR 2 (A Measurement Tool to Assess Systematic Reviews) critical appraisal tool [40]. The tool includes 16 items, each of which measures one domain, with seven critical domains [30]: protocol registration, adequacy of search strategy, justification for excluding studies, risk of bias of included studies, appropriateness of meta-analytical methods, consideration of risk of bias in interpreting results, and assessment of publication bias.

Of the 16 items, eight have dichotomous scores (yes/no); four have ordinal outcome scores (yes, partial yes, no), including whether authors list excluded studies and reasons for exclusion; three ask about quality of meta-analysis methodology (yes, no, no meta-analysis conducted); and one asks about risk of bias with differentiated criteria for RCTs and non-randomized study interventions (NRSI) with three outcome scores (yes, partial yes, no). Of note, this instrument is not designed to produce an overall score. Inter-rater agreement on 224 scored items for the 14 included studies was 82%. The two raters recorded scores and reasons independently for each item, met to discuss all disagreements, and resolved all disagreements by consensus.

### 2.4. Data Extraction

Multiple team members collaborated on the development and testing of data extraction categories and forms. Data were extracted in duplicate by two authors (E.J.T., L.H.-T) using methodology specific to umbrella reviews [41]. The following data were extracted: citation information; study objectives; participant characteristics and number; setting(s) (e.g., home, school); focus of intervention (physical activity, nutrition); description of parent component; contact hours/dose; sources searched; dates, number, country of origin, and study design of included primary studies, including both RCTs and NRSI; quality appraisal instrument and rating; method of analysis; outcome assessed; results (including significance and direction); and heterogeneity. All disagreements were discussed until consensus was reached. Missing or inconsistent data were flagged as such. For example, for two prevention-focused reviews, participant number was not specified or not extractable.

### 2.5. Data Items and Measures

The primary outcome was child weight or weight status. Measures and changes in measures of weight and weight status were recorded: BMI, BMI percentile, BMI-Z scores, body fat, weight, weight gain, weight loss, change in weight status (e.g., overweight to normal weight), % overweight, and % weight change. Descriptive data on parent involvement in the SRs/MAs differentiated parent-only and parent-as-target interventions, parent-child interventions, and family treatment or family-based interventions. If one or more subsets of a review included extractable information on parent involvement and others included parents only as informants on children’s behavior or attributes, wherever possible, information from the subset that included parent involvement was compared with the subset that did not include parent involvement.

### 2.6. Data Synthesis

Data were synthesized narratively at the level of the SR/MA. Reviews were grouped by prevention or treatment focus for synthesis. Because results varied both within and across SRs/MAs as a function of type of parent involvement, comparator, and weight status outcome, review results and synthesis were differentiated according to each of these categories.

## 3. Results

We identified 4158 records through database searches (PRISMA flowchart, Figure 1). Of the 2411 publications that remained after pre-screening, 129 publications were selected for full-text review. Of these, 115 were excluded after full-text review; most were excluded because of the ages of study participants (out of our range of interest, not described, not extractable by age) (*n* = 82) or because they were not a SR/MA (*n* = 18); fifteen studies were excluded for other reasons. Excluded articles and reason for exclusion are listed in Appendix A. Fourteen publications met all criteria outlined above and are included in this umbrella review.

### 3.1. Study Characteristics

Study characteristics are described in Table 1. Six SRs [42,43,44,45,46,47], one MA [48], and seven SRs with MAs [49,50,51,52,53,54,55] discussed 216 unique primary studies with publication years ranging from 1975 to 2018. Primary studies included within each SR/MA are listed in Appendix A. The seven reviews classified as both a SR and MA will be referred to as MA. Eight reviews (three SR, five MA) examined the treatment [42,43,48,50,51,52,53] and six reviews (three SR, three MA) examined the prevention [44,46,47,49,54,55] of child overweight and obesity. Of note, the stated objective of Gori et al. was to determine the “efficacy of interventions aimed at preventing childhood obesity” [49]; however, only children with overweight and obesity were included in the primary studies reviewed. This wording made it unclear whether the focus was prevention, treatment, or both; it is included in the prevention studies based on the authors’ declared objective.

Intervention designs within the 14 SRs/MAs included randomized controlled trials (RCTs), cluster RCTs, quasi-experimental, and pre-/post-test designs. Primary studies were conducted in an array of countries, with the United States, Australia, Switzerland, the Netherlands, Israel, and Germany as frequent study locations. Of all the primary studies included by the 14 SRs/MAs, each primary article was included an average of 1.23 ± 0.50 times across the 14 reviews, with only 8/216 articles cited three times, indicating minimal overlap of primary studies across reviews (see Appendix A). All included studies in this umbrella review explicitly described “parents”, which is the term used henceforth.

The total extractable number of participants was 130,260; 119,299 participants were derived from prevention-focused and 10,961 from treatment-focused SRs/MAs. The actual participant number is higher but not specifiable because of omissions in two prevention-focused reviews [46,49].

### 3.2. Quality Assessment

Table 2 presents attainment by the 14 SRs/MAs of the 16 AMSTAR 2 criteria. Items 11, 12, and 15 apply only to MAs. The seven items regarded by AMSTAR 2 developers as critical for the quality of SRs/MAs are bolded. The criterion most frequently attained was the inclusion of all four PICO elements (Population, Intervention, Control Group, Outcome), which is a non-critical domain. One study [51] attained all seven critical domains (with one partial yes) and one [50] attained six of the seven critical domains (with one partial yes). Two studies [52,55] attained four or five critical domains. Three studies [44,45,48] did not attain any critical domains. Of the eight reviews that employed meta-analytic methods, only four [50,51,53,55] were rated as including appropriate meta-analytic methods (item 11) and only four [51,52,54,55] were rated as assessing publication bias (item 15). The treatment-focused reviews were assessed as attaining more critical domains than prevention-focused reviews.

### 3.3. Interventions and Study Settings

The treatment and prevention approaches described within each of the 14 SRs/MAs were implemented in myriad settings, and intervention techniques were diverse. For the treatment-focused studies, primary care or outpatient settings were common, along with the home environment. With the exception of some primary studies in the review by Gori et al. [49], the prevention-focused primary studies included schools or preschools as a target.

### 3.4. Treatment-Focused Reviews

Eight SRs/MAs examined treatment-focused interventions [42,43,45,48,50,51,52,53]. Sbruzzi et al. included both prevention and treatment-focused primary studies [53]. However, only the treatment-focused studies were included in the current review because a parent component was not described for the prevention studies. Table 3 presents the research question(s) that guided each treatment-focused SR/MA and the application of the question(s) to the current umbrella review. Three general themes or categories of questions characterized the eight treatment-focused SRs/MAs: (1) Are interventions/treatments involving parents more effective than a comparator? (2) Are parent-only interventions better than child-only interventions? and (3) Are parent-only interventions equivalent to parent-child interventions? Five of the eight treatment-focused reviews addressed the question of effectiveness of interventions for children that involved parents [45,48,51,52,53], while three reviews addressed the effectiveness of interventions targeting only parents [42,43,50]. We expected equivalent effectiveness of parent-only and parent-child interventions due to the involvement of the parent in each condition (see Table 3).

#### 3.4.1. Treatment Focus: Parents Involved

Two MAs [48,52] evaluated family behavioral treatment (i.e., behavior change) interventions. One MA by Oude Luttikhuis et al. [52] was assessed in the current study as attaining 5/7 critical quality domains. This review compared family treatment with minimal/standard care controls and revealed a significantly greater decrease in child BMI-z score of −0.06 (95% CI: −0.12 to −0.01) for the family behavioral treatments at six-month follow-up but not upon later follow-up. The MA by Young et al. [48] of family treatments and other treatments was assessed as attaining 0/7 critical AMSTAR 2 quality domains; this MA revealed family treatments resulted in a significant decrease in child percent overweight of −0.62 (95% CI: −0.80 to −0.44) at post-test. The other treatments resulted in a statistically non-significant decrease. Thus, although they used different meta-analytic strategies and were widely divergent in quality, the MAs that evaluated family behavioral treatments [48,52] identified results supporting greater effectiveness of family behavioral treatments compared to other treatments and minimal/standard care controls.

One SR [45] and two MAs [51,53] evaluated treatment interventions involving parents. The SR by McLean et al. [45] attained 0/5 of the critical domains of quality; this review included seven primary studies targeting children’s food intake and/or physical activity. Only 3/7 primary studies compared parent-child interventions with a control group or child-only comparator. Of these three, two led to improved outcomes. The other four studies included in the SR compared various parent/family approaches with other parent/family approaches, and none led to improved outcomes.

The two MAs [51,53] included behavioral interventions as well as interventions targeting nutrition and PA. Sbruzzi et al. [53], which was rated by the authors as attaining 3/7 critical domains for quality, restricted studies to educational treatment interventions with parent involvement that targeted behavior, diet, and/or physical activity. MA of five educational interventions with BMI as an outcome revealed a significant decrease in child BMI of −0.86 kg/m^2^ (95% CI: −1.59 to −0.14), whereas MA of six interventions that measured BMI-z revealed a non-significant decrease in BMI-z of −0.06 (95% CI: −0.16 to 0.03). The other MA by Mead et al. [51] is the most extensive review of treatment-focused studies included in this umbrella review, with 65/70 primary articles having a parent component; this MA also received the highest AMSTAR 2 quality ratings (7/7 domains) in this umbrella review. However, there was no narrative synthesis of the 65 papers beyond a meta-analyses that included 38 of those 65 papers; for this reason, only those 38 primary articles are described in the present umbrella review. The results of those analyses are organized in Table 3 by outcome (change in BMI, BMI-z, or body weight) and type of inclusion of parent (parent as target versus parent involvement). The comparator in all cases was no treatment or usual care. The number of trials with parent as target ranged from one to three depending on the outcome; only one MA of one trial with body weight as outcome was statistically significant. The number of trials that described parent involvement was 13 (change in body weight), 20 (change in BMI), and 32 (change in BMI-z), with primary articles appearing in multiple meta-analyses. MA of parent involvement for each of the three outcomes was significant with Z-statistic values (i.e., test for significance of effect) ranging from 3.20 to 3.36, confirming that treatments targeting diet, PA, and/or behavior of children with overweight or obesity and involving their parents are effective in impacting obesity-related measures.

#### 3.4.2. Treatment Focus: Parent Only

Three reviews evaluated parent-only interventions, two SRs [42,43] and one MA [50]. Both SRs were rated as attaining 2/5 of the critical AMSTAR 2 quality domains. One SR by Jang et al. asked whether interventions that targeted only parents were effective; the other by Ewald et al. asked whether such interventions were effective in comparison to parent-child and child-only interventions. Findings from Jang et al. [43] support that targeting only parents is effective if the comparator is usual care or a waitlist control but not if the comparator is an alternate intervention that involved parents. Ewald et al. [42] evaluated six primary studies (reported in 10 papers); two study protocols also were described by Ewald et al. but are not included in this review because no data were available. Of those six primary studies with data, only one included a parent-only and child-only group that were largely equivalent in all other aspects, with children in the parent-only group showing greater weight loss. The remaining five primary studies compared parent-only and parent-child interventions, with four showing no difference and one revealing a greater reduction in overweight for the parent-only than the parent-child group. Both reviews [42,43] show modest effectiveness of parent involvement, but conclusions about limited effectiveness must be qualified by low review quality. Further, absence of differences between parent-only and parent-child interventions is consistent with the hypothesis that outcomes of parent-only and parent-child interventions should be equivalent because both involve parents.

The MA by Loveman et al. [50] attained 6/7 of the AMSTAR 2 critical quality domains and evaluated whether diet, physical activity, and behavioral interventions delivered only to parents to treat obesity and overweight in children aged 5 to 11 years were effective. MA of parent-only interventions compared to waitlist controls revealed parent-only interventions were effective, with a mean difference in child BMI-z score reduction of −0.12 (95% CI: −0.21 to −0.04) in post-intervention assessments in two trials and a mean difference of −0.10 (95% CI: −0.19 to −0.01) in longest follow-up assessments in two trials. However, MA revealed no evidence that parent-only interventions were better than minimal control interventions for reducing child BMI-z scores. Furthermore, there was evidence that parent-only and parent-child interventions were equivalent because no MA revealed significant differences between these two intervention types. Thus, although diverging widely in quality, the three SRs/MAs summarized in this section [42,43,50] are congruent in identifying parent-only interventions to treat child obesity/overweight as more effective than usual care and waitlist controls and that parent-only and parent-child interventions are generally equivalent. The high quality of the critical AMSTAR 2 ratings of the MA by Loveman et al. [50] and the findings in this MA for (1) greater effectiveness of parent-only interventions versus waitlist controls and (2) equivalent effectiveness of parent-only versus parent-child interventions underscores the importance of parent involvement in diet, physical activity, and behavioral interventions for children with overweight or obesity.

### 3.5. Prevention-Focused Reviews

Table 4 presents the research purpose or question that guided each of the six prevention-focused SRs/MAs followed by application of that purpose or question to the current umbrella review. The synthesis below is guided by two themes: whether interventions with unspecified degree of parent involvement—at home or at school—revealed effective results (two reviews) and whether school-based interventions with parent involvement were effective in preventing child obesity (four reviews).

#### 3.5.1. Prevention Focus: Parent Component—Involvement Not Specified

One review by Gori et al. [49] attained 1/7 critical AMSTAR 2 quality domains and focused on diet and/or physical activity prevention interventions conducted in the home setting or combined home and school setting. Of all MAs in the review (diet and physical activity interventions, alone and in combination; delivered in home, school, or combined setting), only the MA of combined diet and physical activity interventions delivered in combined home and school settings revealed a statistically significant reduction in child BMI-z scores, specifically an effect of −0.15 (CI: −0.22 to −0.07). These results suggest interventions addressing diet and physical activity together and including both home and school are effective in reducing obesity and overweight in school-age children.

Laws et al. [44] attained 0/5 critical AMSTAR 2 quality domains and sought to determine whether preschool interventions were effective in preventing obesity in children from socioeconomically disadvantaged families. The SR reviewed seven primary studies, all including parents; four were evaluations of the Hip Hop to Health intervention. Three primary studies reported statistically significant effects on child BMI or body fat; all studies were high or moderate quality. Four studies reported non-significant findings; three of these were classified as low quality. The association between study quality and significance of findings in this review was striking. Although the results of both Gori et al. and Laws et al. are consistent with the importance of parent involvement for prevention interventions for children with overweight or obesity, the low-quality ratings taper the strength of this conclusion.

#### 3.5.2. Prevention Focus: Parent Component—Involvement Specified

The four prevention SRs/MAs that focused on parent involvement were conducted in preschool and school [46] or school settings [47,54,55]. The SR by Nixon et al. [46] attained 1/5 critical AMSTAR 2 quality domains and reviewed seven interventions of 4- to 6-year-old children with parent involvement; in five other interventions, parents completed informed consent only or informed consent plus questionnaires but did not participate (i.e., without parent involvement). Three of seven parent involvement studies reported statistically significant effects on weight and 4/7 did not, whereas only 1/5 studies without parent involvement showed significant improvement in weight and 4/5 did not. These results support effectiveness of parent involvement.

One SR [47] and two MAs [54,55] reviewed school-based prevention-focused interventions that included parent involvement. The SR by Verjans-Janssen et al. [47] was rated as having attained 2/5 critical quality domains. Of the 18 primary studies within this SR, 11/18 reported reduced BMI/BMI-z results favoring the intervention group. Of these 11 studies, seven reported results favoring the intervention for both outcomes whereas four reported significance for only one outcome or subgroup, for example, effective only for children with overweight or obesity.

The MA by Sobol-Goldberg et al. [54] was rated as having attained 3/7 critical quality domains and reported eight primary studies in which the authors identified parent involvement. MA of three comprehensive, one-year-long studies with parent involvement revealed a statistically significant reduction in child BMI of −0.393 (CI: −0.773 to −0.012). MA of five shorter-duration studies with parent involvement revealed a significant reduction in child BMI of −0.102 (CI: −0.165 to −0.040). In contrast, MA of two comprehensive one-year long interventions that did not include parent involvement reported a non-significant decrease in child BMI of −0.023.

The other MA of school-based prevention interventions by Oosterhoff et al. [55] attained 4/7 critical AMSTAR 2 quality domains and included 83 primary papers of lifestyle interventions, 53 of which featured parent involvement. MA of all 83 RCTs revealed a statistically significant effect of the lifestyle (diet, physical activity, education) interventions but with large heterogeneity (I^2^ = 87.3%). Meta-regression analyses of all 83 RCTs revealed parent involvement was a moderator of lifestyle interventions, significantly reducing child BMI by −0.42 (CI: −0.81 to −0.002).

In sum, each of the three SRs/MAs of school-based prevention interventions [47,54,55] provides evidence of the importance of parent involvement for prevention intervention effects. These three SRs/MAs also received higher ratings on the critical AMSTAR 2 quality domains than the two reviews that involved parents but did not specify the degree/nature of the involvement [44,49]. Thus, the combination of review quality and review findings underscores the conclusion of effectiveness of parent involvement for prevention intervention effects.

**Table 4 nutrients-13-03227-t004:** Research Questions and Results for SRs/MAs of Prevention Interventions.

Author (SR, MA, Both)	Research Question/Purpose	Umbrella Review Research Question(s)	Results	Conclusion
Gori (both) [49]	To update the Waters et al. (2011) meta-analysis results about the effectiveness of educational and lifestyle interventions aimed at preventing child obesity.	Are interventions that include diet or PA components or the combination of diet and PA effective in family settings or family and school settings combined?	**Primary Outcome: BMI-SDS Reduction** **Intervention: Diet alone delivered in family setting or combined family + school: NS** **Intervention: PA alone * or PA + diet in family setting: NS** **Intervention: Combined (diet + physical activity) interventions delivered in combined (family + school) settings** −0.15 (CI: −0.22 to −0.07); Z = 4.02, *p* < 0.0001; I^2^ = 94%; 5 or 6 studies; 451 participants**. Notes: * Table 1 indicates 0 family studies with PA alone. Text on page 241 does not specify whether NS studies in family setting were PA alone or PA + diet. Appendix A lists 2 PA-alone family studies, but descriptions of interventions do not mention PA. ** One of the 6 studies included in Appendix A is not listed as combined study in Appendix A but as a school study. Whether the N of studies is 5 or 6 cannot be determined.	Combined diet and PA interventions delivered to children in settings that combined home and school settings resulted in significant reductions in BMI-SDS.
Laws (SR) [44]	To systematically review the literature to examine effectiveness of interventions to prevent obesity or improve obesity-related behaviors in children aged 0–5 years from Indigenous families or families experiencing socioeconomic disadvantage.	Are interventions delivered in preschool settings effective in preventing obesity in families who experience socioeconomic disadvantage?	**Primary Outcome: BMI or body fat** **3/7 studies found significant intervention effects on BMI (2 studies) or body fat (1 study)—1 high quality study; 2 moderate quality studies.** **4/7 studies did not find significant intervention effects—3 low quality studies and 1 study with quality not reported in the review.** DETAILS Providing feedback to parents and referral of children with overweight/obesity to physician significantly lowered BMI Classroom PA + education + environment changes + parent discussions significantly reduced body fat 4 HipHop studies: study of Black preschool children [56] significantly lowered BMI in intervention children compared to controls effectiveness trial with Black preschool children—BMI- NS study of Latino preschool children [57]—NS study of Latino preschool children with intensive parental component [58]—NS. Healthy and Ready to Learn for Latino children did not significantly affect BMI.	Some evidence for effectiveness of preschool interventions delivered to families who experience socioeconomic disadvantage as long as studies are of moderate or high quality.
Nixon (SR) [46]	To identify the most effective behavioral models and strategies underpinning preschool- and school-based interventions for preventing obesity in 4–6 year olds.	Are interventions for 4 to 6 year olds with parental involvement effective in preventing obesity?	**Primary Outcome: Weight Status** **3/7 studies with parental involvement found significant impact on weight status; 4/7 with parental involvement did not find significant impact on weight status.** In one study the outcome was significant for the African American cohort but not for the Latino cohort. **1/5 studies with no parent involvement reported a significant impact on weight status.** Parents completed informed consent only or informed consent plus questionnaires but did not participate in interventions.	Some evidence for effectiveness of obesity prevention interventions with parental involvement.
Oosterhoff (both) [55]	To systematically review the evidence of the impact of school-based lifestyle RCTS on children’s BMI and blood pressure.	Lifestyle RCTs with parent involvement will reduce child BMI	**Primary Outcome: BMI** 53/83 unique RCTs included a parental involvement component in addition to the single or multiple lifestyle components; k = 151 effect sizes. Three-level model of impact of school-based lifestyle RCTs was significant with large heterogeneity: −0.072 (CI: −0.106 to −0.038), *p* < 0.001, I^2^ = 87.3% k = 151; RCTs = 83. Parent involvement was a significant moderator of lifestyle RCTS with involvement increasing the positive effects of school-based lifestyle interventions: −0.42 (CI: −0.81 to −0.002), *p* < 0.05.	Parent involvement significantly enhanced the positive impact of school-based lifestyle RCTs in reducing child BMI.
Sobol-Goldberg (both) [54]	To evaluate efficacy of school-based obesity prevention programs. To test the hypothesis that studies that were comprehensive and at least one year long with parental support would have the best results.	Do school-based interventions that address nutrition and physical activity and include parent involvement reduce child BMI?	** 8 studies of children with parent involvement identified by the authors **** Primary Outcome: BMI ** **MA of comprehensive, 1-year-long studies with parental involvement** − 0.393 (CI: −0.773 to −0.012); *p* < 0.05; 3 studies, 3579 participants **MA of comprehensive, shorter duration studies with parental involvement** − 0.102 (CI: −0.165 to −0.040); *p* < 0.01; 5 studies, 4131 participants There were 0 comprehensive shorter duration studies without parental involvement. There were 9 studies that were “none of the above.” These would appear to be studies that were not comprehensive. The authors did not classify these 9 studies as to parental involvement; therefore, data could not be extracted.	The five studies identified as being comprehensive and with parental involvement resulted in significant reductions in child BMI.
Verjans-Janssen (SR) [47]	To study effectiveness of school-based physical activity and nutrition interventions with direct parental involvement on children’s BMI or BMI z-score, physical activity, sedentary behavior and nutrition behavior.	Do school-based physical activity and nutrition interventions with direct parental involvement reduce BMI and/or BMI-z?	** 18 studies of children aged 5 to 12 on average, all with direct parental involvement, and BMI or BMI-z scores as primary outcome ** 11/18 studies with results favoring the intervention group 7/18 studies with all results for BMI and BMI-z favoring intervention group: o 5 reported small effect sizes o 1 reported a moderate effect size and 1 reported a large effect size 4/18 studies had mixed results: effective for BMI but not BMI-z; effective for children classified as normal and overweight only; effective for children with obesity and overweight only; effective for boys only.	61% of school-based PA and nutrition interventions with direct parental involvement reduced child BMI or BMI-z.

## 4. Discussion

This umbrella review is the first to examine both treatment and prevention meta-analyses and systematic reviews of child obesity interventions that describe a parent component. When all 14 SRs/MAs included in this umbrella review are considered together, the inclusion of a parent component appears beneficial for child obesity-related outcomes, including BMI, BMI-Z score, and weight, among children aged 3–12 years old. This effect was observed for SRs/MAs focused on either treatment or prevention of overweight and obesity. Further, our selection criteria yielded large, diverse samples of individuals from many different countries, a multitude of study settings, and an array of intervention approaches. The emergence of parent involvement as a positive factor across the diversity of settings and approaches suggests the importance of considering this factor for childhood obesity interventions.

For treatment interventions, effectiveness of parent involvement was reported across widely divergent types of treatments, from family therapy and behavior change programs [52] to interventions across multiple implementation settings including homes, primary care, schools, university research clinics, and community settings [51]. Our findings align with and extend those of recent umbrella reviews that reported parent involvement was positively associated with weight management for children ≤ 18 years of age [30] and in parent-targeted or parent-child treatment interventions [29].

We also extend previous findings with the inclusion of both treatment- and prevention-focused reviews. Prevention interventions were less widely divergent in setting, with most occurring in schools. Effectiveness was demonstrated for parent involvement in prevention interventions that were more comprehensive [54] as well as in a comparison of 53 interventions with parent involvement against 30 interventions without involvement; the results, which showed that parent involvement significantly enhanced effectiveness of school-based interventions [55]. These findings suggest parent involvement can be included in interventions for children aged 3–12 years old beyond the family home environment; this result supports findings from previous umbrella reviews about the importance of parent involvement in school-based prevention and treatment interventions for adolescents and children [31] and for promoting healthy eating within child care settings [24]. Other umbrella reviews have demonstrated a positive influence of parent involvement with interventions focused on single behaviors, such as physical activity [59] or gardening [60]. Findings from the present umbrella review suggest that parent involvement in multi-level and combination interventions, such as those that include both diet and physical activity approaches, may have the greatest impact on child weight-related outcomes.

The authors assessed quality of all 14 SRs/MAs with AMSTAR 2 and focused on the seven domains regarded as critical [40]. Of the 14 reviews, three were rated as very low in quality (two treatment [45,48] and one prevention review [44]). Although parent involvement was supported within these low-quality reviews, comparators and intervention effects were not consistent, or wide divergence of intervention components (e.g., referral to physician, parent discussions, classroom intervention) rendered conclusions inappropriate. In contrast, two reviews were rated as very high; both were MAs of treatment interventions. One [51] found effectiveness of parent involvement in dietary and physical activity interventions across all outcomes measured, with significant changes in BMI, BMI-z, and weight across more than 6000 participants combined from included primary studies. The other review [50] also reported effectiveness of parent involvement, but the effectiveness varied by the type of comparator (e.g., waitlist control versus minimal contact control) for parent only interventions. Importantly, both of these high-quality reviews described the low quality of the component primary studies, underscoring the continued need for better quality of obesity treatment interventions. Additionally noteworthy is that in all reviews that compared parent-only and parent-child interventions, the two interventions were equally effective. The fact that these two types of interventions were effective across multiple reviews [42,43,50] underscores the importance of parent involvement.

Many aspects of the parent–child relationship and the home environment can impact obesity risk for children. For example, children’s dietary intake is influenced not only by parents’ dietary choices but also by parenting style [13,61,62]. As such, interventions that do not address underlying parenting styles likely would not be successful [63]. Additionally, physical and social environments are important contributors to children’s eating patterns, and parents’ behaviors, attitudes, and feeding styles are known to contribute to the social food environment [12]. Similarly, parents impact physical activity behaviors of children; studies show that adult and child physical activity levels are correlated [58] and that parenting practices and styles influence child physical activity [56]. Moreover, the caregiver–child relationship appears important for health behaviors in the childcare setting [57], which is important for the children who spend significant amounts of time in this environment. More research is needed in interventions to better define and evaluate optimal parent and caregiver involvement strategies, which ranged widely in the present umbrella review.

### Limitations

We focused on children aged 3–12 years old given the developmental differences between infants, toddlers, children, and adolescents and for reasons related to the transition into preschool programs around age three and into the teenage years after age 12. For many children < 5 years old, a significant amount of time may be spent in a childcare setting. However, several childcare-based studies were excluded from the present review because data for children aged 3 and older were not extractable. While broadening the age criteria might have included additional studies, this increase would have diminished precision of conclusions and recommendations for parents of children aged 3–12 years old. In the present umbrella review, we also were unable to determine the type and duration of parent involvement for all reviews (i.e., directly participating in behavior interventions versus indirectly receiving education via newsletters) or the quality of the parent relationship with the child. For example, a parent may be directly involved in an intervention, but the quality of the parent–child relationship may be poor, which may moderate effects of parent involvement. Finally, our careful comparison of primary studies across all reviews (see Appendix A) showed that while there is very little overlap, 6/8 studies in the SR of Jang et al. [43] were also included in the 20 primary studies by Loveman et al. [50]. This is the only evidence of significant overlap across the SRs/MAs we included, and it serves to underscore that the overlap is a limitation but also that it is an exception.

## 5. Conclusions

Parent involvement appears to be a beneficial component of nutrition- and physical activity-focused interventions for the prevention and treatment of overweight and obesity among children aged 3–12 years old. This effectiveness was demonstrated for multiple types of parent involvement in treatment interventions, including family-based approaches, parent-only interventions, and parent-child interventions, which may allow for greater flexibility for intervention planning and delivery. Findings from prevention reviews highlight the importance of parent involvement in interventions that include the school environment. The emergence of parent support as an important component across a wide range of prevention and treatment approaches and within multiple settings provides an impetus for future research to investigate the most effective methods of involving parents, including type and duration of involvement.

## Figures and Tables

**Figure 1 nutrients-13-03227-f001:**
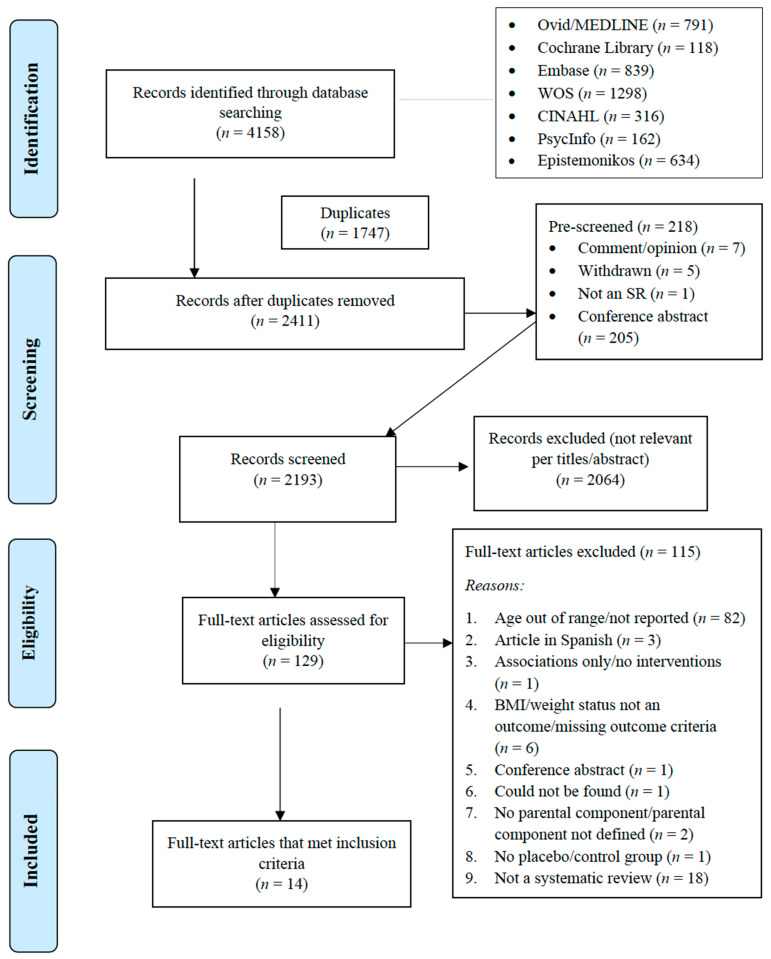
Flowchart of the process of literature search and extraction of studies meeting the inclusion criteria.

**Table 1 nutrients-13-03227-t001:** Characteristics of Included Meta-Analysis and Systematic Reviews by Intervention Focus (Treatment and Prevention).

Treatment Focused											
Author, Year ^a^	Year Range of Primary Studies	Number of Primary Studies	Participant Number	Participant Characteristics	Study Setting (s)	Country of Origin of Included Studies ^b^	Intervention Targets Included		Target Group (Comparators)	Study Design of Primary Studies	Meta-Analysis Conducted
							**Nutr**	**PA**			
Ewald, 2014	1998–2011	8 studies reported in 10 papers and 2 protocols (12 papers in total)	466	Children aged 5–14 years (stated objective to target 5–12 years); children with overweight or obesity at baseline	Family based, university, hospital/outpatient, community	Israel, USA, Switzerland, Australia	✓	✓	Parent-only (parent-child, child-only)	RCTs	No
Jang, 2015	2007–2014	7 studies reported in 8 papers	765 ^c^	Children aged 3–13 years (mean age < 10 in 6/7 studies); majority non-Hispanic white (when reported); children with overweight or obesity at baseline	Community, hospital, university, not specified	USA, Australia, Netherlands, Belgium	✓	✓	Parent-only (usual care, active control, alterative or partial intervention)	RCTs	No
Loveman, 2015	1975–2015	20 studies reported in 20 main papers for qualitative synthesis; of these, 14 studies were included for the meta-analysis	3057	Children aged 2–13 years (mean 4.9–11.5 years; stated objective to target 5–11 y); majority non-Hispanic White; children with overweight or obesity at baseline	Outpatient, community, university, primary care, combination	USA, Australia, Netherlands, Israel, Switzerland, Iran, Belgium	✓	✓	Parent-only (parent-child, waitlist control, minimal contact, other parent-only)	RCTs with at least six months of outcome assessment	Yes
McLean, 2003 *	1981–1994	7 studies described in 11 papers	300	Children aged 6–13	Not specified	USA, Sweden	✓	✓	Parent-child (comparators not specified)	Randomized trials	No
Mead, 2017 *	1984–2016	Analysis included 38 papers that described parental involvement (*n* = 35) or targeted parents (*n* = 3)	4150	Children aged 6–12 years Children with overweight, obesity, or severe obesity at baseline	Outpatient, primary care, home, community, hospital, school	USA, UK, Germany, Spain, Australia, Israel, Sweden, New Zealand, Italy, Mexico, Canada, Finland, Malaysia	✓	✓	Parent-child, parent-only (true control, usual/standard care, other parent-child)	RCTs with at least six months of follow-up	Yes
Oude Luttikhuis, 2009 *	2006–2008	8 studies	708 randomized (579 completed)	Children aged 5–12 years; children with overweight or obesity at baseline	Outpatient, school	Australia, Sweden, Israel, UK, Finland, Switzerland, unknown	✓	✓	Parent-only, parent-child (parent-child, other parent-only, waitlist control, usual/standard care)	RCTs	Yes
Sbruzzi, 2013 *^,d^	2007–2011	8 studies	849	Children aged 6–12 years; children with overweight or obesity at baseline	Not specified	USA, UK, Australia, Finland, Sweden, Malaysia	✓	✓	Parent-only, parent-child, family-based (wait list control, usual/standard care, minimal contact)	RCTs	Yes
Young, 2007	1982–2004	16 studies	666	Children aged 5–13 years (stated objective to target 5–12 years); children with overweight or obesity at baseline	Not specified	Not specified	✓	✓	Parent as “helper” or parent treated Concurrently (other treatment, control)	Not reported	Yes
Treatment Participants			10,961								
**Prevention Focused**											
Gori, 2017 *	2003–2014	Mismatch in number of studies across published text, Table 1	Not Extractable	Children aged 6–12 years Children with overweight or obesity at baseline	Family or combined family and school based	USA, Australia, Argentina, Italy, Netherlands, Israel, France, Spain, UK	✓	✓	Family, parent-child, not specified (usual/standard care, minimal contact, true control, not specified)	RCT, non-RCT	Yes
Laws, 2014 *	2005–2013	7 studies	4294	Children aged 2.5–6 years Studies targeted preschools with high numbers of children with migrant, Black, Latino, or low-income backgrounds	Preschools	USA, France, Switzerland	✓	✓	Child-parent-teacher (comparators not specified)	Cluster RCT, quasi-experimental	No
Nixon, 2012 *	1998–2010	7 studies that included parental involvement	Not Specified	Children aged 4–6.9 years	Preschools and schools	Germany, Greece, Scotland, Switzerland, USA, Australia, China, England, New Zealand, Thailand	✓	✓	Parent-child, not specified (comparators not specified)	RCTs, non-RCTs	No
Oosterhoff, 2016 *	1985–2013	83 studies reported in 89 papers; 53 studies (in 54 papers) for which a parental component was described. All 83 studies were included in multivariable meta-regression model for BMI	72,934 ^e^	Children aged 4–12 years Range of BMIs at baseline; only 3 studies targeted children with overweight/obesity	Schools	Europe, North America, Oceania, South America, North Africa (individual countries not listed)	✓	✓	Target group not specified (control groups received no intervention beyond typical school-based activities per inclusion criteria)	RCTs	Yes
Sobol-Goldberg, 2013 *	Not Extractable	8 studies for the target age range for which parental involvement is described	7710	Children aged 5–12 years Range of BMIs at baseline	Schools	Not extractable	✓	✓	Target group not specified (control groups received no intervention per inclusion criteria)	RCTs	Yes
Verjans-Janssen, 2018 *	1999–2018	18 studies that included BMI/BMI z-score as an outcome	34,361	Children aged 4–12 years	Schools	China, USA, Australia, Greece, Chile, Germany, Italy, Mexico	✓	✓	Parent-child; all studies had direct parent involvement (comparators not specified)	RCT, quasi-experimental, pre-/post-test	No
Prevention Participants			119,299								
Total Reported Participants			130,260 ^f^								

^a^ Authors marked with an ‘*’ denote papers where an extractable subset of studies that met the inclusion criteria of the present study are reported rather than all studies included in the referenced publication. ^b^ Country of origin is listed in order from most to least, where reported. ^c^ Participants in this study identified as families. ^d^ Sbruzzi also reported and separately analyzed prevention-focused studies; however, a parental component was not described for these studies, and the prevention-focused section is therefore not included in the current umbrella review. ^e^ Participant number reflects 85 total studies, two of which did not report on BMI but rather on blood pressure outcomes only (another study objective); of these, 83 were included in the BMI meta-analysis. ^f^ Participant number reflects the number that could be counted and does not include those studies for which the number of participants was not reported or extractable; as such, this number is lower than the actual number of participants.

**Table 2 nutrients-13-03227-t002:** AMSTAR-2 Assessment of Included Systematic Reviews and Meta-Analyses.

	AMSTAR-2 Item Number															
Review (Year), Treatment or Prevention Focus	1	2	3	4	5	6	7	8	9	10	11	12	13	14	15	16
Ewald (2014), T	Y	**N**	N	**PY**	N	N	**N**	PY	**Y**	N	**N/A**	N/A	**N**	N	**N/A**	N
Gori (2017), P	Y	**N**	Y	**N**	N	N	**N**	Y	**N**	N	**N**	Y	**Y**	Y	**N**	Y
Jang (2015), T	Y	**N**	N	**N**	N	Y	**N**	Y	**Y**	N	**N/A**	N/A	**Y**	N	**N/A**	N
Laws (2014), P	Y	**N**	Y	**N**	N	N	**N**	N	**N**	N	**N/A**	N/A	**N**	N	**N/A**	Y
Loveman (2015), T	Y	**PY**	Y	**Y**	Y	Y	**Y**	Y	**Y**	Y	**Y**	Y	**Y**	Y	**N**	Y
McLean (2003), T	Y	**N**	N	**N**	N	N	**N**	N	**N**	N	**N/A**	N/A	**N**	N	**N/A**	N
Mead (2017), T	Y	**PY**	N	**Y**	Y	Y	**Y**	Y	**Y**	Y	**Y**	Y	**Y**	Y	**Y**	Y
Nixon (2012), P	N	**N**	N	**N**	Y	N	**Y**	N	**N**	N	**N/A**	N/A	**N**	N	**N/A**	Y
Oosterhoff (2016), P	Y	**N**	N	**N**	Y	N	**N**	N	**PY**	N	**Y**	Y	**Y**	Y	**Y**	Y
Oude Luttikhuis (2009), T	Y	**PY**	Y	**PY**	Y	Y	**Y**	Y	**Y**	Y	**N**	N	**N**	N	**Y**	Y
Sbruzzi (2013), T	Y	**N**	N	**PY**	Y	Y	**N**	PY	**PY**	N	**Y**	N	**N**	Y	**N**	Y
Sobol-Goldberg (2013), P	Y	**PY**	Y	**N**	N	Y	**N**	N	**PY**	N	**N**	N	**N**	N	**Y**	Y
Verjans-Janssen (2018), P	Y	**N**	Y	**N**	Y	N	**N**	N	**PY**	N	**N/A**	N/A	**Y**	Y	**N/A**	Y
Young (2007), T	N	**N**	N	**N**	Y	N	**N**	N	**N**	N	**N**	N	**N**	Y	**N**	N

P = prevention; T = treatment; Y = yes; N = no; PY = partial yes; N/A = not applicable. 1 = PICO Elements; 2 = Prior Protocol; 3 = Study Designs; 4 = Search Strategy; 5 = Study Selection; 6 = Data Extraction; 7 = Excluded Studies; 8 = PICO Details; 9 = Risk of Bias Assessment; 10 = Funding Sources; 11 = Meta-Analysis Methods; 12 = Risk of Bias Impact on Results; 13 = Risk of Bias Discussion; 14 = Explain Heterogeneity; 15 = Publication Bias; 16 = Conflict of Interest. Bold text indicates items designated as the seven critical domains by the AMSTAR 2 developers. See citation [40] for a full description of items.

**Table 3 nutrients-13-03227-t003:** Research Questions and Results for SRs/MAs of Treatment Interventions.

Author (SR, MA, Both)	Research Question/Purpose	Umbrella Review Research Question (s)	Results	Conclusion
Ewald (SR) [42]	Are parent-only interventions effective treatments of obesity in children aged 5–12 years compared with child-only or parent-child interventions?	Are parent-only (PO) interventions better than child-only (CO) interventions?	1/6 studies offered semi-equivalent PO vs. CO comparison and PO group showed significantly greater weight loss than CO.	PO may be better than CO.
		Are parent-only (PO) interventions equivalent to parent-child (PC) interventions?	4/6 studies revealed NS difference in weight status between PO and PC interventions. 1/6 studies revealed greater change in overweight for PO than PC intervention group.	PO and PC appear to be equivalent (4/6 versus 1/6).
Jang (SR) [43]	To evaluate interventions for child overweight and obesity that target parents.	Are interventions targeting parents and focused on children’s healthy eating (HE) and physical activity (PA) effective?	5/7 studies that compared intervention group(s) to either a usual care group or a waitlist control group (WLC) revealed significant decreases in BMI or BMI z-scores. 2/7 studies that compared alternative interventions to the focal intervention did not reveal significant between-groups differences in BMI z-scores.	Interventions targeting parents and promoting child HE and PA are more effective than usual care or WLC.
Loveman (both) [50]	To assess the efficacy of diet, physical activity, and behavioral interventions delivered only to parents to treat obesity and overweight in children aged 5 to 11 years.	Are parent-only (PO) interventions better than wait list control conditions (WLC) and minimal contact control interventions (MCI)?	**MA of PO versus WLC** **Change in BMI z score** Post-intervention mean difference in BMI z score of −0.12 (95% CI: −0.21 to −0.04); Z = 2.95, *p* = 0.003; I^2^ = 0.0%; 2 trials; 153 participants; low-quality evidence. Longest follow-up mean difference in BMI z score of −0.10 (95% CI: −0.19 to −0.01); Z = 2.09, *p* = 0.04; I^2^ = 0.0%; 136 participants; 2 trials with 3 treatment arms (one unique from above); low-quality evidence. **BMI percentile and BMI** Only two single studies (separate publications) with poor methodological quality and not analyzed by MA. **MA of PO versus MCI** **Change in BMI z score** MA of two comparisons at post-intervention within one study revealed a mean difference of −0.00 between PO and MCI (95% CI: −0.08 to 0.08); Z = 0.01, *p* = 0.99; I^2^ = 0.0%; 170 participants; 1 trial with 3 treatment arms; low-quality evidence. MA of two comparisons at longest follow-up within one study revealed a mean difference of 0.01 between PO and MCI (95% CI: −0.07 to 0.09); Z = 0.24, *p* = 0.81; I^2^ = 0.0%; 165 participants; 1 trial with 3 treatment arms (same trial as for post-intervention); low-quality evidence. **BMI percentile and BMI** Four trials of BMI percentile at post-intervention could not be combined for MA because standardization was lacking. None revealed significant treatment effects. One trial examined BMI at post-intervention with no difference between groups. One trial examined change in BMI percentile at follow-up with NS results. MA of two trials of BMI change at longest follow-up revealed a mean difference of −0.12 between PO and MCI (95% CI: −0.39 to 0.15); Z = 0.86, *p* = 0.39; I^2^ = 0.0%; 614 participants; 2 trials; low-quality evidence.	There is evidence that PO interventions are better than WLC for reducing BMI Z scores.
				There is no evidence that PO interventions are better than MCI interventions for reducing BMI Z scores, BMI percentile, and BMI.
		Are parent-only (PO) interventions equivalent to parent-child (PC) interventions?	**MA of PO versus PC** **Change in BMI z score** Post-intervention mean difference in BMI z score of −0.06 (95% CI: −0.13 to 0.02); Z = 1.49, *p* = 0.14; I^2^ = 37%; 277 participants; 3 trials with 4 treatment groups, low-quality evidence. Longest follow-up mean difference in BMI z score of −0.04 (95% CI: −0.15 to 0.08); Z = 0.59, *p* = 0.56; I^2^ = 38%; 267 participants; 3 trials with 4 treatment groups (same trials as post-intervention above), low-quality evidence 2 trials not analyzed due to missing SD; one reported PO significantly better. **% Overweight** Two trials not analyzed by MA: one reported significantly greater decrease for PO than PC at both post-intervention and longest follow-up; the other reported NS differences in decreases at both post-intervention and longest follow-up.	There is evidence that PO interventions and PC interventions are equivalent because no MA of PO versus PC revealed significant differences.
		Are parent-only (PO) interventions equivalent to other parent only (OPO) interventions?	**MA of PO versus OPO** **Change in BMI z score (No MA)** Five trials were not analyzed by MA due to no consistency in interventions or comparators across trials. 4/5 reported NS findings. 1/5 reported increasing PA and decreasing sedentary activity were each significantly better than growth monitoring. **BMI percentile and BMI (No MA)** 2/2 studies (one BMI; one BMI percentile) reported NS differences between PO and OPO.	There is evidence that PO interventions are equivalent to OPO interventions because only 1/7 studies revealed a significant difference.
McLean (SR) [45]	To identify trials evaluating family involvement in weight control, weight maintenance, and weight loss interventions targeting food intake and/or physical activity.	Did trials involving parents lead to weight control or weight loss?	**Findings at post-intervention time point (2/7 led to improved outcomes; 5/7 did not)** **Trials comparing parent-child with child-only or parent-child with control group** Targeting parent and child resulted in more reduction in % overweight than targeting child alone. Family-based treatment resulted in greater decrease in % over BMI than yoked controls (NS at 24 months). Child vs. parent-child condition resulted in NS difference in % child overweight. **Trials comparing various parent or family approaches** Targeting parent control vs. child self-control revealed NS difference in % weight change. Family therapy vs. conventional treatment revealed NS difference in weight control. Adding parent training to behavioral weight reduction resulted in NS decrease in overweight from baseline to follow-up. Behavioral weight reduction alone resulted in significant decrease in child overweight status. Enhanced child involvement vs. standard treatment resulted in NS difference in % child overweight status.	There is some evidence from trials comparing parent-child or family interventions with child-only interventions or controls.
				There is no evidence that other types of trials led to weight control or weight loss.
Mead (both) [51]	How effective are diet, physical activity and behavioral interventions in reducing the weight of children aged 6 to 11 years with overweight or obesity?	Are diet, PA, and behavioral interventions that include parental involvement more effective than no treatment/usual care?	**Findings at final follow-up; analyzed studies classified as low-quality evidence** **Change in BMI—Parental Involvement** Mean difference in BMI of −0.65 (95% CI: −1.04 to −0.25); Z = 3.2, *p* = 0.0007; I^2^ = 69.91%; 20 trials; 2217 participants; low-quality evidence. **Change in BMI—Parent-Targeted** Mean difference in BMI of 0 (95% CI: −0.81 to 0.81); 1 trial, 146 participants. Tests for heterogeneity and effect were not applicable. **Change in BMI-Z—Parental Involvement** Mean difference in BMI-Z of −0.07 (95% CI: −0.11 to −0.03); Z = 3.25, *p* = 0.0006, I^2^ = 60.44%; 32 trials; 2927 participants. **Change in BMI-Z—Parent-Targeted** Mean difference in BMI-Z of 0.01 (95% CI: −0.06 to 0.08); Z = 0.22; *p* = 0.83, I^2^ = 0%, 3 trials; 748 participants. **Change in body weight—Parental Involvement** Mean difference in body weight of −1.32 (95% CI: −2.09 to −0.55); Z = 3.36, *p* = 0.0004, I^2^ = 0%, 13 trials; 1273 participants. **Change in body weight—Parent-Targeted** Mean difference in BMI-Z of −2 (95% CI: −3.02 to −0.98); Z = 3.83, *p* = 0, I^2^ omitted; 1 trial; 79 participants.	Diet, PA, and behavioral interventions that include parental involvement are more effective than no treatment/usual care for every outcome evaluated: Change in BMI; Change in BMI-Z; Change in body weight.
Oude Luttikhuis (both) [52]	To assess the efficacy of any combination of lifestyle (dietary, physical activity, behavioral therapy), drug or surgical interventions, compared with any other combination of these interventions or no treatment in children and adolescents.	Are behavioral family programs for treatment of childhood obesity better than standard or minimal care? Note: The authors switched the intervention and control groups of an included primary study (Golan 2006) to maintain consistency with other included studies so that parent-child was designated as the intervention group and parent-only as the control.	**Behavioral interventions for families/parents and children; no report of cross-study quality** **Change in BMI Z score** Only 8/24 studies met MA criterion for analyses to be based on intention-to-treat principles. **MA of family programs versus minimal or standard care at 6 months or first assessment after 6 months** 4/8 studies met all criteria for MA which revealed an effect of −0.06 (95% CI: −0.12 to −0.01), Z = 2.18, *p* = 0.03; 301 participants, I^2^ = 61%. **Other 4 interventions for families/parents and children at 6 months** ¾ studies revealed intervention and comparison both decreased and/or NS difference. Fourth study revealed smaller increase in BMI for school-based family treatment than conventional therapy but not compared to untreated control group. **MA of behavioral family programs compared to minimal or standard care at 12 or 24 months follow up** 3/7 studies met all criteria for MA which revealed an effect of −0.04 (95% CI: −0.12 to 0.04), Z = 0.91, *p* = 0.36; 264 participants, I^2^ = 0.0%. **Other 4 interventions for families/parents and children at 12 or 24 months follow up** For study comparing school-based family treatment to conventional therapy or untreated control group, smaller increase in BMI for school-based family treatment than control group but no longer for conventional therapy. Other studies were reported as having results persist from end of intervention to 12 or 24 months.	MA provides some evidence that parent/family programs are better than standard or minimal care.
Sbruzzi (both) [53]	To systematically review educational interventions, including behavioral modification, nutrition and physical activity, as compared to usual care or no intervention, for prevention or treatment of obesity in school children. ^1^	Did treatment trials involving parents lead to decreased obesity compared to usual care or no intervention?	**MA of treatment versus usual care or no intervention** **Change in BMI** MA of five studies revealed a significant reduction in BMI of −0.86 kg/m^2^ (95% CI: −1.59 to −0.14), *p* = 0.02; I^2^ = 51%, low-quality evidence. **Change in BMI z score** MA of six studies revealed a NS reduction in BMI z score of −0.06 (95% CI: −0.16 to 0.03), *p* = 0.16, I^2^ = 37%, very low-quality evidence.	Treatment of obesity with behavior modification, nutrition, and/or physical activity leads to reduction in BMI.
Young (MA) [48]	To determine the effectiveness of family-based treatments for weight loss in children.	Are family-behavioral treatments (FBT) more effective than other treatments without parent involvement (OT)?Are FBT more effective than control conditions (CC)?	**NOTE: I^2^ not reported. Quality not reported.** **Decrease in % overweight at post-test** 16 FBT treatments (after removal of 1 study responsible for heterogeneity) resulted in a significant decrease, d = −0.62, SD = 0.10 (95% CI: −0.80 to −0.44). 3 OT resulted in a non-significant decrease, d = −0.52, SD = 0.41 (95% CI: −1.49 to 0.44). 5 CC resulted in a non-significant decrease d = −0.18, SD = 0.47 (95% CI: −0.75 to 0.39). T-test for difference between OT and FBT approached significance, t(20) = 2.41, *p* = 0.052 (95% CI: 0.79 to 11.14). No t-test was reported for comparison of CC with FBT. **Change in weight (pounds) at post-test** 6 FBT treatments resulted in a significant decrease, d = −0.61, SD = 0.46 (95% CI: −1.10 to −0.12). 2 OT resulted in a non-significant decrease, d =−0.35, SD = 0.54 (95% CI: −4.90 to 4.20) 2 CC resulted in a non-significant increase, d = 0.46, SD = 0.27 (95% CI: −3.65 to 4.57) No t-test for difference was conducted for any comparison. **Decrease in BMI or BMI-z at post-test** Due to too few FBT studies for MA, individual treatment effects per study were reported. For FBT on BMI: 1 study reported small decrease in BMI; the other reported small increase in BMI. For FBT on BMI-z: the only study reported a large negative effect. **Decrease in % overweight, weight (pounds), BMI, and BMI-z at follow-up** FBT treatments (number of studies unspecified) resulted in significant decrease in % overweight but OT and CC data insufficient to compute Hedges *d* for comparison. 2 FBT treatments and one OT reported Hedges *d* for pounds. FBT combined *d* was NS; measurement periods for FBT and OT were different so groups not combined in MA.	Family-behavioral treatments are more effective than other treatments without parental involvement. Family-behavioral treatments are more effective than control conditions.

## Data Availability

The data used and analyzed during the current study are available from the corresponding author upon reasonable request.

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
