# Peer review of "Parent Involvement in Diet or Physical Activity Interventions to Treat or Prevent Childhood Obesity: An Umbrella Review"

_nutrients, 2021, doi:10.3390/nu13093227_

Round 1

Reviewer 1 Report

A very interesting paper. 

Author Response

We are very appreciative of your time spent reviewing this paper. We are not sure if the line "Extensive editing of English language and style required" was checked in error given that no other comments were provided. Regardless, we have completed a thorough check of the document for any spelling or grammar errors.

Reviewer 2 Report

This paper describes an umbrella review, which aimed to synthesize evidence on effects of parent involvement in diet and physical activity treatment and prevention interventions on obesity risk among children aged 3-12 years. Fourteen systematic reviews and/or meta-analysis were included for the final analysis. This is the only umbrella review that investigates this aim.  Their findings support inclusion of a parent component in both treatment and prevention interventions to improve child weight/weight status outcomes. It is well conducted, will full information provided, the search is included, clear in and exclusion criteria, a study protocol has been published and the search and analysis are repeatable with all the information proved. Furthermore, it follows the PRISMA guidelines. To conclude this is a well-written umbrella review, with all necessary information provided in a organised and structured manner.

I have only one comment. Please also use People-first language in the tables. It is already used in the text of the manuscript, just not fully in the tables yet.

Author Response

Thank you for your time in reviewing this paper. We appreciate you noting the inconsistency in using people-first language in the tables, which we have updated to reflect your recommendation.

Reviewer 3 Report

This comprehensive umbrella review use state of the art methods to search and evaluate included reviews and meta-analyses. 

I have no further comments that could improve this work significantly. 

Author Response

Thank you very much for the time spent reviewing our manuscript.